# Self-Supervised Time Series Representation Learning by Inter-Intra Relational Reasoning

## Abstract

Self-supervised learning achieves superior performance in many domains by extracting useful representations from the unlabeled data. However, most of traditional self-supervised methods mainly focus on exploring the inter-sample structure while less efforts have been concentrated on the underlying intra-temporal structure, which is important for time series data. In this paper, we present **Self-Time**: a general **Self**-supervised **Time** series representation learning framework, by exploring the inter-sample relation and intra-temporal relation of time series to learn the underlying structure feature on the unlabeled time series. Specifically, we first generate the inter-sample relation by sampling positive and negative samples of a given anchor sample, and intra-temporal relation by sampling time pieces from this anchor. Then, based on the sampled relation, a shared feature extraction backbone combined with two separate relation reasoning heads are employed to quantify the relationships of the sample pairs for inter-sample relation reasoning, and the relationships of the time piece pairs for intra-temporal relation reasoning, respectively. Finally, the useful representations of time series are extracted from the backbone under the supervision of relation reasoning heads. Experimental results on multiple real-world time series datasets for time series classification task demonstrate the effectiveness of the proposed method. Code and data are publicly available [1].

## 1 Introduction

Time series data is ubiquitous and there has been significant progress for time series analysis (Das, 1994) in machine learning, signal processing, and other related areas, with many real-world applications such as healthcare (Stevner et al., 2019), industrial diagnosis (Kang et al., 2015), and financial forecasting (Sen et al., 2019).

Deep learning models have emerged as successful models for time series analysis (Hochreiter & Schmidhuber, 1997; Graves et al., 2013; Shukla & Marlin, 2019; Fortuin et al., 2019; Oreshkin et al., 2020). Despite their fair share of success, the existing deep supervised models are not suitable for high-dimensional time series data with a limited amount of training samples as those data-driven approaches rely on finding ground truth for supervision, where data labeling is a labor-intensive and time-consuming process, and sometimes impossible for time series data. One solution is to learn useful representations from unlabeled data, which can substantially reduce dependence on costly manual annotation.

Self-supervised learning aims to capture the most informative properties from the underlying structure of unlabeled data through the self-generated supervisory signal to learn generalized representations. Recently, self-supervised learning has attracted more and more attention in computer vision by designing different pretext tasks on image data such as solving jigsaw puzzles (Noroozi & Favaro, 2016), inpainting (Pathak et al., 2016), rotation prediction(Gidaris et al., 2018), and contrastive learning of visual representations(Chen et al., 2020), and on video data such as object tracking (Wang & Gupta, 2015), and pace prediction (Wang et al., 2020). Although some video-based ap-

---

[1] Anonymous repository link.

Figure 1: Exploring inter-sample relation, and multi-scale intra-temporal relation of time series. Here, an example of 3-scale temporal relations including short-term, middle-term and long-term temporal relation is given for illustration.

proaches attempt to capture temporal information in the designed pretext task, time series is far different structural data compared with video. More recently, in the time series analysis domain, some metric learning based self-supervised methods such as triplet loss (Franceschi et al., 2019) and contrastive loss (Schneider et al., 2019; Saeed et al., 2020), or multi-task learning based self-supervised methods that predict different handcrafted features (Pascual et al., 2019a; Ravanelli et al., 2020) and different signal transformations (Saeed et al., 2019; Sarkar & Etemad, 2020) have emerged. However, few of those works consider the intra-temporal structure of time series. Therefore, how to design an efficient pretext task in a self-supervised manner for time series representation learning is still an open problem.

In this work, we present SelfTime: a general self-supervised time series representation learning framework. Inspired by relational discovery during self-supervised human learning, which attempts to discover new knowledge by reasoning the relation among entities (Goldwater et al., 2018; Patacchiola & Storkey, 2020), we explore the inter-sample relation reasoning and intra-temporal relation reasoning of time series to capture the underlying structure pattern of the unlabeled time series data. Specifically, as shown in Figure 1, for inter-sample relation reasoning, given an anchor sample, we generate from its transformation counterpart and another individual sample as the positive and negative samples respectively. For intra-temporal relation reasoning, we firstly generate an anchor piece, then, several reference pieces are sampled to construct different scales of temporal relation between the anchor piece and the reference piece, where relation scales are determined based on the temporal distance. Note that in Figure 1, we only show an example of 3-scale temporal relations including short-term, middle-term, and long-term relation for an illustration, whereas in different scenarios, there could be different temporal relation scale candidates. Based on the sampled relation, a shared feature extraction backbone combined with two separate relation reasoning heads are employed to quantify the relationships between the sample pairs or the time piece pairs for inter-sample relation reasoning or intra-temporal relation reasoning, respectively. Finally, the useful representations of time series are extracted from the backbone under the supervision of relation reasoning heads on the unlabeled data. Overall, SelfTime is simple yet effective by conducting the designed pretext tasks directly on the original input signals.

Our main contributions are three-fold: (1) we present a general self-supervised time series representation learning framework by investigating different levels of relations of time series data including inter-sample relation and intra-temporal relation. (2) We design a simple and effective intra-temporal relation sampling strategy to capture the underlying temporal patterns of time series. (3) We conduct extensive experiments on different categories of real-world time series data, and systematically study the impact of different data augmentation strategies and temporal relation sampling strategies on self-supervised learning of time series. By comparing with multiple state-of-the-art baselines, experimental results show that SelfTime builds new state-of-the-art on self-supervised time series representation learning.

## 2 RELATED WORK

**Time Series Modeling.** In the last decades, time series modeling has been paid close attention with numerous efficient methods, including distance-based methods, feature-based methods, ensemble-based methods, and deep learning based methods. Distance-based methods (Berndt & Clifford,

1994; Górecki & Łuczak, 2014) try to measure the similarity between time series using Euclidean distance or Dynamic Time Warping distance, and then conduct classification based on 1-NN classifiers. Feature-based methods aim to extract useful feature for time series representation. Two typical types including bag-of-feature based methods (Baydogan et al., 2013; Schäfer, 2015) and shapelet based methods (Ye & Keogh, 2009; Hills et al., 2014). Ensemble-based methods (Lines & Bagnall, 2015; Bagnall et al., 2015) aims at combining multiple classifiers for higher classification performance. More recently, deep learning based methods (Karim et al., 2017; Ma et al., 2019; Cheng et al., 2020) conduct classification by cascading the feature extractor and classifier based on MLP, RNN, and CNN in an end-to-end manner. Our approach focuses instead on self-supervised representation learning of time series on unlabeled data, exploiting inter-sample relation and intra-temporal relation of time series to guide the generation of useful feature.

**Relational Reasoning.** Reasoning the relations between entities and their properties makes significant sense to generally intelligent behavior (Kemp & Tenenbaum, 2008). In the past decades, there has been an extensive researches about relational reasoning and its applications including knowledge base (Socher et al., 2013), question answering (Johnson et al., 2017; Santoro et al., 2017), video action recognition (Zhou et al., 2018), reinforcement learning (Zambaldi et al., 2019), and graph representation (Battaglia et al., 2018), which perform relational reasoning directly on the constructed sets or graphs that explicitly represent the target entities and their relations. Different from those previous works that attempt to learn a relation reasoning head for a special task, inter-sample relation reasoning based on unlabeled image data is employed in (Patacchiola & Storkey, 2020) to learn useful visual representation in the underlying backbone. Inspired by this, in our work, we focus on time series data by exploring both inter-sample and intra-temporal relation for time series representation in a self-supervised scenario.

**Self-supervised Learning.** Self-supervised learning has attracted lots of attention recently in different domains including computer vision, audio/speech processing, and time series analysis. For image data, the pretext tasks including solving jigsaw puzzles (Noroozi & Favaro, 2016), rotation prediction (Gidaris et al., 2018), and visual contrastive learning (Chen et al., 2020) are designed for self-supervised visual representation. For video data, the pretext tasks such as frame order validation (Misra et al., 2016; Wei et al., 2018), and video pace prediction (Wang et al., 2020) are designed which considering additional temporal signal of video. Different from video signal that includes plenty of raw feature in both spatial and temporal dimension, time series is far different structural data with less raw features at each time point. For time series data such as audio and ECG, the metric learning based methods such as triplet loss (Franceschi et al., 2019) and contrastive loss (Schneider et al., 2019; Saeed et al., 2020), or multi-task learning based methods that predict different handcrafted features such as MFCCs, prosody, and waveform (Pascual et al., 2019a; Ravanelli et al., 2020), and different transformations of raw signal (Sarkar & Etemad, 2020; Saeed et al., 2019) have emerged recently. However, few of those works consider the intra-temporal structure of time series. Therefore, how to design an efficient self-supervised pretext task to capture the underlying structure of time series is still an open problem.

## 3 METHOD

Given an unlabeled time series set $\mathcal{T} = \{t_n\}_{n=1}^N$, where each time series $t_n = (t_{n,1}, ... t_{n,T})^{\mathrm{T}}$ contains $T$ ordered real values. We aim to learn a useful representation $z_n = f_\theta(t_n)$ from the backbone encoder $f_\theta(\cdot)$ where $\theta$ is the learnable weights of the neural networks. The architecture of the proposed SelfTime is shown in Figure 2, which consists of an inter-sample relational reasoning branch and an intra-temporal relational reasoning branch. Firstly, taking the original time series signals and their sampled time pieces as the inputs, a shared backbone encoder $f_\theta(\cdot)$ extracts time series feature and time piece feature to aggregate the inter-sample relation feature and intra-temporal relation feature respectively, and then feeds them to two separate relation reasoning heads $r_\mu(\cdot)$ and $r_\varphi(\cdot)$ to reason the final relation score of inter-sample relation and intra-temporal relation.

### 3.1 INTER-SAMPLE RELATION REASONING

Formally, given any two different time series samples $t_m$ and $t_n$ from $\mathcal{T}$, we randomly generate two sets of $K$ augmentations $\mathcal{A}(t_m) = \{t_m^{(i)}\}_{i=1}^K$ and $\mathcal{A}(t_n) = \{t_n^{(i)}\}_{i=1}^K$, where $t_m^{(i)}$ and $t_n^{(i)}$ are the $i$-th

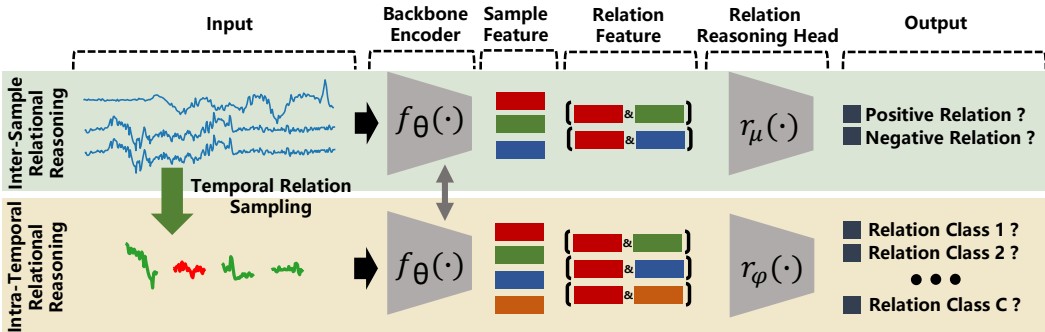

Figure 2: Architecture of SelfTime.

augmentations of $\boldsymbol{t}_m$ and $\boldsymbol{t}_n$ respectively. Then, we construct two types of relation pairs: positive relation pairs and negative relation pairs. A positive relation pair is $(\boldsymbol{t}_m^{(i)}, \boldsymbol{t}_m^{(j)})$ sampled from the same augmentation set $\mathcal{A}(\boldsymbol{t}_m)$, while a negative relation pair is $(\boldsymbol{t}_m^{(i)}, \boldsymbol{t}_n^{(j)})$ sampled from different augmentation sets $\mathcal{A}(\boldsymbol{t}_m)$ and $\mathcal{A}(\boldsymbol{t}_n)$. Based on the sampled relation pairs, we use the backbone encoder $f_\theta$ to learn the relation representation as follows: Firstly, we extract sample representations $\boldsymbol{z}_m^{(i)} = f_\theta(\boldsymbol{t}_m^{(i)})$, $\boldsymbol{z}_m^{(j)} = f_\theta(\boldsymbol{t}_m^{(j)})$, and $\boldsymbol{z}_n^{(j)} = f_\theta(\boldsymbol{t}_n^{(j)})$. Then, we construct the positive relation representation $[\boldsymbol{z}_m^{(i)}, \boldsymbol{z}_m^{(j)}]$, and the negative relation representation $[\boldsymbol{z}_m^{(i)}, \boldsymbol{z}_n^{(j)}]$, where $[\cdot, \cdot]$ denotes the vector concatenation operation. Next, the inter-sample relation reasoning head $r_\mu(\cdot)$ takes the generated relation representation as input to reason the final relation score $h_{2m-1}^{(i,j)} = r_\mu([\boldsymbol{z}_m^{(i)}, \boldsymbol{z}_m^{(j)}])$ for positive relation and $h_{2m}^{(i,j)} = r_\mu([\boldsymbol{z}_m^{(i)}, \boldsymbol{z}_n^{(j)}])$ for negative relation, respectively. Finally, the inter-sample relation reasoning task is formulated as a binary classification task and the model is trained with binary cross-entropy loss $\mathcal{L}_{inter}$ as follows:

$$\mathcal{L}_{inter} = -\sum_{n=1}^{2N}\sum_{i=1}^{K}\sum_{j=1}^{K}(y_n^{(i,j)} \cdot \log(h_n^{(i,j)}) + (1 - y_n^{(i,j)}) \cdot \log(1 - h_n^{(i,j)})) \qquad (1)$$

where $y_n^{(i,j)} = 1$ for the positive relation and $y_n^{(i,j)} = 0$ for the negative relation.

### 3.2 INTRA-TEMPORAL RELATION REASONING

To capture the underlying temporal structure along the time dimension, we try to explore the intra-temporal relation among time pieces and ask the model to predict the different types of temporal relation. Formally, given a time series sample $\boldsymbol{t}_n = (t_{n,1}, ...t_{n,T})^{\mathrm{T}}$, we define an $L$-length time piece $\boldsymbol{p}_{n,u}$ of $\boldsymbol{t}_n$ starting at time step $u$ as a contiguous subsequence $\boldsymbol{p}_{n,u} = (t_{n,u}, t_{n,u+1}, ..., t_{n,u+L-1})^{\mathrm{T}}$. Firstly, we sample different types of temporal relation among time pieces as follows: Randomly sample two $L$-length pieces $\boldsymbol{p}_{n,u}$ and $\boldsymbol{p}_{n,v}$ of $\boldsymbol{t}_n$ starting at time step $u$ and time step $v$ respectively. Then, the temporal relation between $\boldsymbol{p}_{n,u}$ and $\boldsymbol{p}_{n,v}$ is assigned based on their temporal distance $d_{u,v}$, e.g., for similarity, we define the temporal distance $d_{u,v} = |u - v|$ as the absolute value of the difference between two starting step $u$ and $v$. Next, we define $C$ types of temporal relations for each pair of pieces based on their temporal distance, e.g., for similarity, we firstly set a distance threshold as $D = \lfloor T/C \rfloor$, and then, if the distance $d_{u,v}$ of a piece pair is less than $D$, we assign the relation label as 0, if $d_{u,v}$ is greater than $D$ and less than $2D$, we assign the relation label as 1, and so on until we sample $C$ types of temporal relations. The details of the intra-temporal relation sampling algorithm are shown in Algorithm 1.

Based on the sampled time pieces and their temporal relations, we use the shared backbone encoder $f_\theta$ to extract the representations of time pieces firstly, where $\boldsymbol{z}_{n,u} = f_\theta(\boldsymbol{p}_{n,u})$ and $\boldsymbol{z}_{n,v} = f_\theta(\boldsymbol{p}_{n,v})$. Then, we construct the temporal relation representation as $[\boldsymbol{z}_{n,u}, \boldsymbol{z}_{n,v}]$. Next, the intra-temporal relation reasoning head $r_\varphi(\cdot)$ takes the relation representation as input to reason the final relation score $h_n^{(u,v)} = r_\varphi([\boldsymbol{z}_{n,u}, \boldsymbol{z}_{n,v}])$. Finally, the intra-temporal relation reasoning task is formulated as a multi-class classification problem and the model is trained with cross-entropy loss $\mathcal{L}_{intra}$ as

follows:

$$\mathcal{L}_{intra} = -\sum_{n=1}^{N} y_n^{(u,v)} \cdot \log \frac{\exp(h_n^{(u,v)})}{\sum_{c=1}^{C} \exp(h_n^{(u,v)})} \quad (2)$$

By jointly optimizing the inter-sample relation reasoning objective (Eq. 1) and intra-temporal relation reasoning objective (Eq. 2), the final training loss is defined as follows:

$$\mathcal{L} = \mathcal{L}_{inter} + \mathcal{L}_{intra} \quad (3)$$

An overview for training SelfTime is given in Algorithm 2 in Appendix A. SelfTime is an efficient algorithm compared with the traditional contrastive learning models such as SimCLR. The complexity of SimCLR is $O(N^2 K^2)$, while the complexity of SelfTime is $O(NK^2) + O(NK)$, where $O(NK^2)$ is the complexity of inter-sample relation reasoning module, and $O(NK)$ is the complexity of intra-temporal relation reasoning module. It can be seen that SimCLR scales quadratically in both training size $N$ and augmentation number $K$. However, in SelfTime, inter-sample relation reasoning module scales quadratically with the number of augmentations $K$, and linearly with the training size $N$, and intra-temporal relation reasoning module scales linearly with both augmentations and training size.

---

**Algorithm 1: Temporal Relation Sampling.**

**Require:**
  $t_n$: A $T$-length time series.
  $p_{n,u}, p_{n,v}$: two $L$-length pieces of $t_n$.
  $C$: Number of relation classes.
**Ensure:**
  $y_n^{(u,v)} \in \{1, 2, ..., C\}$: The label of the temporal relation between $p_{n,u}$ and $p_{n,v}$.
1: $d_{u,v} = |u - v|, D = \lfloor T/C \rfloor$
2: **if** $d_{u,v} \leq D$ **then**
3:    $y_n^{(u,v)} = 0$
4: **else if** $d_{u,v} \leq 2 * D$ **then**
5:    $y_n^{(u,v)} = 1$
6:    ...
7: **else if** $d_{u,v} \leq (C - 1) * D$ **then**
8:    $y_n^{(u,v)} = C - 2$
9: **else**
10:    $y_n^{(u,v)} = C - 1$
11: **end if**
12: **return** $y_n^{(u,v)}$

---

## 4 EXPERIMENTS

### 4.1 EXPERIMENTAL SETUP

**Datasets.** To evaluate the effectiveness of the proposed method, in the experiment, we use three categories time series including four public datasets CricketX, UWaveGestureLibraryAll (UGLA), DodgerLoopDay (DLD), and InsectWingbeatSound (IWS) from the *UCR Time Series Archive*[2] (Dau et al., 2018), along with two real-world

| Category | Dataset | Sample | Length | Class |
|----------|---------|--------|--------|-------|
| Motion | CricketX | 780 | 300 | 12 |
| | UWaveGestureLibraryAll | 4478 | 945 | 8 |
| Sensor | DodgerLoopDay | 158 | 288 | 7 |
| | InsectWingbeatSound | 2200 | 256 | 11 |
| Device | MFPT | 2574 | 1024 | 15 |
| | XJTU | 1920 | 1024 | 15 |

Table 1: Statistics of Datasets.

bearing datasets XJTU[3] and MFPT[4] (Zhao et al., 2020). All six datasets consist of various numbers of instances, signal lengths, and number of classes. The statistics of the datasets are shown in Table 1.

**Time Series Augmentation** The data augmentations for time series are generally based on random transformation in two domains (Iwana & Uchida, 2020): magnitude domain and time domain. In the magnitude domain, transformations are performed on the values of time series where the values at each time step are modified but the time steps are constant. The common magnitude domain based augmentations include jittering, scaling, magnitude warping (Um et al., 2017), and cutout (DeVries & Taylor, 2017). In the time domain, transformations are performed along the time axis that the elements of the time series are displaced to different time steps than the original sequence. The common time domain based augmentations include time warping (Um et al., 2017), window slicing, and window warping (Le Guennec et al., 2016). More visualization details of different augmentations are shown in Figure 3.

---

[2]https://www.cs.ucr.edu/~eamonn/time_series_data_2018/
[3]https://biaowang.tech/xjtu-sy-bearing-datasets/
[4]https://www.mfpt.org/fault-data-sets/

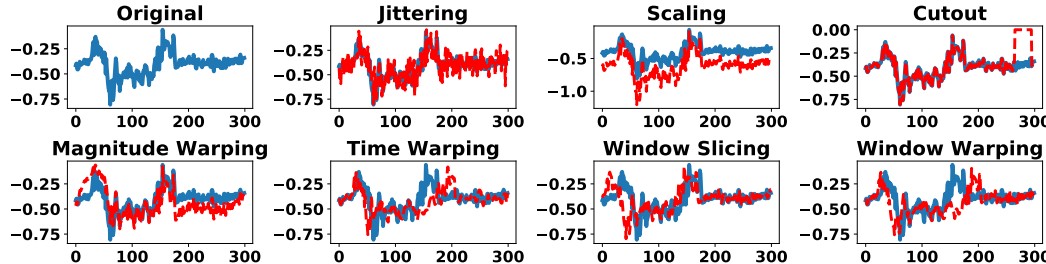

Figure 3: Data augmentations examples from CricketX dataset. The **blue** solid line is the original signal and the **red** dotted lines are the transformations.

**Baselines.** We compare SelfTime against several state-of-the-art methods of self-supervised representation learning:

- *Supervised* consists of a backbone encoder as the same with SelfTime and a linear classifier, which conducts fully supervised training over the whole networks.

- *Random Weights* is the same as *Supervised* in the architecture, but freezing the backbone's weights during the training and optimizing only the linear classifier.

- *Triplet Loss* (Franceschi et al., 2019) is an unsupervised time series representation learning model that uses triplet loss to push a subsequence of time series close to its context and distant from a randomly chosen time series.

- *Deep InfoMax* (Hjelm et al., 2019) is a framework of unsupervised representation learning by maximizing mutual information between the input and output of a feature encoder from the local and global perspectives.

- *Forecast* (Jawed et al., 2020) is a semi-supervised time series classification model that leverages features learned from the self-supervised forecasting task on unlabeled data. In the experiment, we throw away the supervised classification branch and use only the forecasting branch to learn the representations of time series.

- *Transformation* (Sarkar & Etemad, 2020) is a self-supervised model by designing transformation recognition of different time series transformations as pretext task.

- *SimCLR* (Chen et al., 2020) is a simple but effective framework for self-supervised representation learning by maximizing agreement between different views of augmentation from the same sample via a contrastive loss in the latent space.

- *Relation* (Patacchiola & Storkey, 2020) is relational reasoning based self-supervised representation learning model by reasoning the relations between views of the sample objects as positive, and reasoning the relations between different objects as negative.

**Evaluation.** As a common evaluation protocol, linear evaluation is used in the experiment by training a linear classifier on top of the representations learned from different self-supervised models to evaluate the quality of the learned embeddings. For data splitting, we set the training/validation/test split as 50%/25%/25%. During the pretraining stage, we randomly split the data 5 times with different seeds, and train the backbone on them. During the linear evaluation, we train the linear classifier 10 times on each split data, and the best model on the validation dataset was used for testing. Finally, we report the classification accuracy as mean with the standard deviation across all trials.

**Implementation.** All experiments were performed using PyTorch (v1.4.0) (Paszke et al., 2019). A simple 4-layer 1D convolutional neural network with ReLU activation and batch normalization (Ioffe & Szegedy, 2015) were used as the backbone encoder $f_\theta$ for SelfTime and all other baselines, and use two separated 2-layer fully-connected networks with 256 hidden-dimensions as the inter-sample relation reasoning head $r_\mu$ and intra-temporal relation reasoning head $r_\varphi$ respectively (see Table 4 in Appendix B for details). Adam optimizer (Kingma & Ba, 2015) was used with a learning rate of 0.01 for pretraining and 0.5 for linear evaluation. The batch size is set as 128 for all models. For fair comparison, we generate $K = 16$ augmentations for each sample although

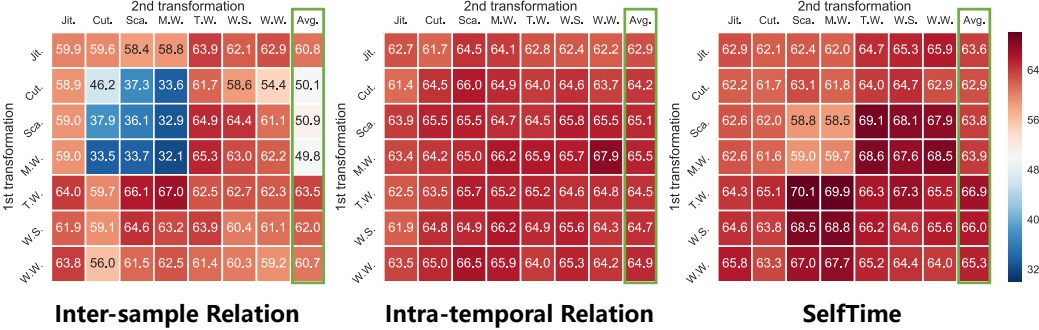

Figure 5: Linear evaluation on CricketX under individual or composition of data augmentations. For all columns but the last, diagonal entries correspond to single transformation, and off-diagonals correspond to composition of two transformations (applied sequentially). The last column reflects the average over the row.

more augmentation results in better performance (Chen et al., 2020; Patacchiola & Storkey, 2020). More implement details of baselines are shown in Appendix D. More experimental results about the impact of augmentation number $K$ are shown in Appendix E.

## 4.2 ABLATION STUDIES

In this section, we firstly investigate the impact of different temporal relation sampling settings on intra-temporal relation reasoning. Then, we explore the effectiveness of inter-sample relation reasoning, intra-temporal relation reasoning, and their combination (SelfTime), under different time series augmentation strategies. Experimental results show that both inter-sample relation reasoning and intra-temporal relation reasoning achieve remarkable performance, which helps the network to learn more discriminating features of time series.

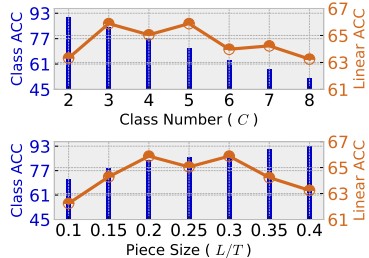

Figure 4: Impact of different temporal relation class numbers and piece sizes on CricketX dataset.

**Temporal Relation Sampling.** To investigate the different settings of temporal relation sampling strategy on the impact of linear evaluation performance, in the experiment, we set different numbers of temporal relation class $C$ and time piece length $L$. Specifically, to investigate the impact of class number, we firstly set the piece length $L = 0.2 * T$ as 20% of the original time series length, then, we vary $C$ from 2 to 8 during the temporal relation sampling. As shown in Figure 4, we show the results of parameter sensitivity experiments on CricketX, where blue bar indicates class reasoning accuracy on training data (Class ACC) and brown line indicates the linear evaluation accuracy on test data (Linear ACC). With the increase of class number, the Linear ACC keeps increasing until $C = 5$, and we find that a small value $C = 2$ and a big value $C = 8$ result in worse performance. One possible reason behind this is that the increase of class number drops the Class ACC and makes the relation reasoning task too difficult for the network to learn useful representation. Similarly, when set the class number $C = 3$ and vary the piece length $L$ from $0.1 * T$ to $0.4 * T$, we find that the Linear ACC grows up with the increase of piece size until $L = 0.3 * T$, and also, either small value or big value of $L$ will drop the evaluation performance, which makes the relation reasoning task too simple (with high Class ACC) or too difficult (with low Class ACC) and prevents the network from learning useful semantic representation. Therefore, as consistent with the observations of self-supervised studies in other domains (Pascual et al., 2019b; Wang et al., 2020), an appropriate pretext task designing is crucial for the self-supervised time series representation learning. In the experiment, to select a moderately difficult pretext task for different datasets, we set {class number ($C$), piece size ($L/T$)} as {3, 0.2} for CricketX, {4, 0.2} for UWaveGestureLibraryAll, {5, 0.35} for DodgerLoopDay, {6, 0.4} for InsectWingbeatSound, {4, 0.2} for MFPT, and {4, 0.2} for XJTU. More experimental results on other five datasets for parameter sensitivity analysis are shown in Appendix E.

| Method | Dataset | | | | | |
|---|---|---|---|---|---|---|
| | CricketX | UGLA | DLD | IWS | MFPT | XJTU |
| Supervised | 62.44±1.53 | 87.83±0.32 | 37.05±1.61 | 66.23±0.45 | 80.29±0.8 | 95.9±0.42 |
| Random Weights | 36.9±0.92 | 70.01±1.68 | 32.95±2.57 | 52.85±1.36 | 46.68±2.35 | 52.58±4.67 |
| Triplet Loss (Franceschi et al., 2019) | 40.01±2.64 | 71.41±1.1 | 41.37±2.47 | 53.61±2.82 | 47.87±2.97 | 53.31±3.43 |
| Deep InfoMax (Hjelm et al., 2019) | 49.16±3.03 | 73.88±2.37 | 38.95±2.47 | 55.99±1.31 | 58.99±2.72 | 76.27±1.83 |
| Forecast (Jawed et al., 2020) | 44.59±1.09 | 75.7±0.9 | 38.74±3.05 | 54.89±1.99 | 52.6±1.65 | 62.28±2.55 |
| Transformation (Sarkar & Etemad, 2020) | 52.12±2.02 | 75.41±0.27 | 35.47±1.56 | 59.68±1.2 | 60.33±3.29 | 85.08±2.01 |
| SimCLR (Chen et al., 2020) | 59.0±3.19 | 74.9±0.92 | 37.74±3.8 | 56.19±0.98 | 71.81±1.21 | 88.84±0.63 |
| Relation (Patacchiola & Storkey, 2020) | 65.3±0.43 | 80.87±0.78 | 42.84±3.23 | 62.0±1.49 | 73.53±0.65 | 95.14±0.72 |
| SelfTime (ours) | **68.6±0.66** | **84.97±0.83** | **49.1±2.93** | **66.87±0.71** | **78.48±0.94** | **96.73±0.76** |

Table 2: Linear evaluation of representations learned by different models on different datasets.

**Impact of Different Relation Modules and Data Augmentations.** To explore the effectiveness of different relation reasoning modules including inter-sample relation reasoning, intra-temporal relation reasoning, and their combination (SelfTime), in the experiment, we systematically investigate the different data augmentations on the impact of linear evaluation for different modules. Here, we consider several common augmentations including magnitude domain based transformations such as jittering (Jit.), cutout (Cut.), scaling (Sca.), magnitude warping (M.W.), and time domain based transformations such as time warping (T.W.), window slicing (W.S.), window warping (W.W.). Figure 5 shows linear evaluation results on CricketX dataset under individual and composition of transformations for inter-sample relation reasoning, intra-temporal relation reasoning, and their combination (SelfTime). Firstly, we observe that the composition of different data augmentations is crucial for learning useful representations. For example, inter-sample relation reasoning is more sensitive to the augmentations, and performs worse under Cut., Sca., and M.W. augmentations, while intra-temporal relation reasoning is less sensitive to the manner of augmentations, although it performs better under the time domain based transformation. Secondly, by combining both the inter-sample and intra-temporal relation reasoning, the proposed SelfTime achieves better performance, which demonstrates the effectiveness of considering different levels of relation for time series representation learning. Thirdly, we find that the composition from a magnitude-based transformation (e.g. scaling, magnitude warping) and a time-based transformation (e.g. time warping, window slicing) facilitates the model to learn more useful representations. Therefore, in this paper, we select the composition of magnitude warping and time warping augmentations for all experiments. Similar experimental conclusions also hold for other datasets. More experimental results on the other five datasets for evaluation of the impact of different relation modules and data augmentations are shown in Appendix F.

## 4.3 TIME SERIES CLASSIFICATION

In this section, we evaluate the proposed method by comparing with other state-of-the-arts on time series classification task. Firstly, we conduct linear evaluation to assess the quality of the learned representations. Then, we evaluate the performance of all methods in transfer learning by training on the unlabeled source dataset and conduct linear evaluation on the labeled target dataset. Finally, we qualitatively evaluate and verify the semantic consistency of the learned representations.

**Linear Evaluation.** Following the previous studies (Chen et al., 2020; Patacchiola & Storkey, 2020), we train the backbone encoder for 400 epochs on the unlabeled training set, and then train a linear classifier for 400 epochs on top of the backbone features (the backbone weights are frozen without back-propagation). As shown in Table 2, our proposed SelfTime consistently outperforms all baselines across all datasets. SelfTime improves the accuracy over the best baseline (Relation) by 5.05% (CricketX), 5.06% (UGLA), 14.61% (DLD), 7.85% (IWS), 6.73% (MFPT), and 1.67% (XJTU) respectively. Among those baselines, either global features (Deep InfoMax, Transformation, SimCLR, Relation) or local features (Triplet Loss, Deep InfoMax, Forecast) are considered during representation learning, they neglect the essential temporal information of time series except Triplet Loss and Forecast. However, by simply forecasting future time pieces, Forecast cannot capture useful temporal structure effectively, which results in low-quality representations. Also, in Triplet Loss, a time-based negative sampling is used to capture the inter-sample temporal relation among time pieces sampled from the different time series, which is cannot directly and efficiently capture the intra-sample temporal pattern of time series. Different from all those baselines, Self-Time not only extracts global and local features by taking the whole time series and its time pieces

| Method | Source→Target | | | | | |
|---|---|---|---|---|---|---|
| | UGLA→CricketX | CricketX→UGLA | IWS→DLD | DLD→IWS | XJTU→MFPT | MFPT→XJTU |
| Supervised | 31.31±2.76 | 71.85±1.2 | 22.9±2.55 | 44.31±3.25 | 63.15±2.08 | 82.58±3.98 |
| Random Weights | 36.9±0.92 | 70.01±1.68 | 32.95±2.57 | 52.85±1.36 | 46.68±2.35 | 52.58±4.67 |
| Triplet Loss (Franceschi et al., 2019) | 30.08±4.66 | 55.32±2.51 | 34.67±3.12 | 45.22±3.09 | 53.75±2.96 | 59.24±3.02 |
| Deep InfoMax (Hjelm et al., 2019) | 45.92±2.3 | 64.2±4.19 | 37.42±1.99 | 47.75±1.74 | 56.75±0.77 | 77.14±3.14 |
| Forecast (Jawed et al., 2020) | 32.67±0.86 | 72.42±1.17 | 25.47±2.93 | 55.39±1.36 | 53.08±2.75 | 61.74±3.92 |
| Transformation (Sarkar & Etemad, 2020) | 39.24±2.25 | 70.4±1.98 | 30.0±1.66 | **57.71±0.83** | 54.71±1.68 | 69.81±8.03 |
| SimCLR (Chen et al., 2020) | 45.48±3.46 | 65.67±2.91 | 36.21±1.02 | 36.2±4.03 | 63.11±2.0 | 81.62±3.95 |
| Relation (Patacchiola & Storkey, 2020) | 52.55±2.67 | 75.67±0.54 | 36.0±1.52 | 56.29±1.82 | 70.27±1.14 | 92.77±1.15 |
| SelfTime (ours) | **55.04±2.58** | **77.77±0.35** | **45.0±1.48** | 57.8±1.33 | **75.06±1.84** | **93.79±2.46** |

Table 3: Domain transfer evaluation by training with self-supervision on the unlabeled source data and linear evaluation on the labeled target data (e.g. source→target: UGLA→CricketX).

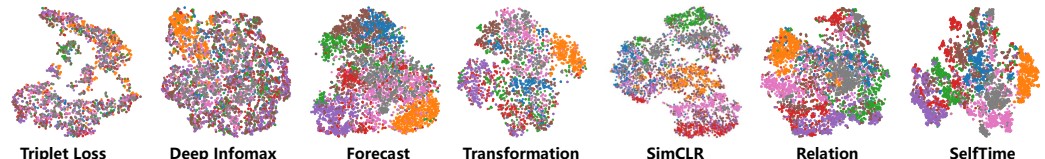

**Triplet Loss**   **Deep Infomax**   **Forecast**   **Transformation**   **SimCLR**   **Relation**   **SelfTime**

Figure 6: t-SNE visualization of the learned feature on UGLA dataset. Different colors indicate different labels.

as inputs during feature extraction, but also captures the implicit temporal structure by reasoning intra-temporal relation among time pieces.

**Domain Transfer.** To evaluate the transferability of the learned representations, we conduct experiments in transfer learning by training on the unlabeled source dataset and conduct linear evaluation on the labeled target dataset. In the experiment, we select two datasets from the same category as the source and target respectively. As shown in Table 3, experimental results show that our SelfTime outperforms all the other baselines under different conditions. For example, SelfTime achieves an improvement over the Relation by 4.73% on UGLA→CricketX transfer, and over Deep InfoMax 20.2% on IWS→DLD transfer, and over Relation 6.81% on XJTU→MFPT transfer, respectively, which demonstrates the good transferability of the proposed method.

**Visualization.** To qualitatively evaluate the learned representations, we use the trained backbone to extract the features and visualize them in 2D space using t-SNE (Maaten & Hinton, 2008) to verify the semantic consistency of the learned representations. Figure 6 shows the visualization results of features from the baselines and the proposed SelfTime on UGLA dataset. It is obvious that by capturing global sample structure and local temporal structure, SelfTime learns more semantic representations and results in better clustering ability for time series data, where more semantic consistency is preserved in the learned representations by our proposed method.

## 5 CONCLUSION

We presented a self-supervised approach for time series representation learning, which aims to extract useful feature from the unlabeled time series. By exploring the inter-sample relation and intra-temporal relation, SelfTime is able to capture the underlying useful structure of time series. Our main finding is that designing appropriate pretext tasks from both the global-sample structure and local-temporal structure perspectives is crucial for time series representation learning, and this finding motivates further thinking of how to better leverage the underlying structure of time series. Our experiments on multiple real-world datasets show that our proposed method consistently outperforms the state-of-the-art self-supervised representation learning models, and establishes a new state-of-the-art in self-supervised time series classification. Future directions of research include exploring more effective intra-temporal structure (i.e. reasoning temporal relation under the time point level), and extending the SelfTime to multivariate time series by considering the causal relationship among variables.

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

| | Layer Description | Output Tensor Dim. |
|---|---|---|
| #0 | Input time series (or time piece) | $1 \times T$ (or $1 \times L$) |
| **Backbone Encoder** | | |
| #1 | Conv1D(1, 8, 4, 2, 1)+BatchNorm+ReLU | $8 \times T/2$ (or $8 \times L/2$) |
| #2 | Conv1D(8, 16, 4, 2, 1)+BatchNorm+ReLU | $16 \times T/4$ (or $16 \times L/4$) |
| #3 | Conv1D(16, 32, 4, 2, 1)+BatchNorm+ReLU | $32 \times T/8$ (or $32 \times L/8$) |
| #4 | Conv1D(32, 64, 4, 2, 1)+BatchNorm+ReLU +AvgPool1D+Flatten+Normalize | 64 |
| **Inter-Sample Relation Reasoning Head** | | |
| #1 | Linear+BatchNorm+LeakyReLU | 256 |
| #2 | Linear+Sigmoid | 1 |
| **Intra-Temporal Relation Reasoning Head** | | |
| #1 | Linear+BatchNorm+LeakyReLU | 256 |
| #2 | Linear+Softmax | $C$ |

Table 4: Implementation detail of SelfTime. Here, we denote 1D convolutional layer as Conv1D(in_channels, out_channels, kernel_size, stride, padding).

Jiangliu Wang, Jianbo Jiao, and Yun-Hui Liu. Self-supervised video representation learning by pace prediction. In *European conference on computer vision, ECCV*, 2020.

Xiaolong Wang and Abhinav Gupta. Unsupervised learning of visual representations using videos. In *Proceedings of the IEEE international conference on computer vision, ICCV*, pp. 2794–2802, 2015.

Donglai Wei, Joseph J Lim, Andrew Zisserman, and William T Freeman. Learning and using the arrow of time. In *Proceedings of the IEEE Conference on Computer Vision and Pattern Recognition, CVPR*, pp. 8052–8060, 2018.

Lexiang Ye and Eamonn Keogh. Time series shapelets: a new primitive for data mining. In *Proceedings of the 15th ACM SIGKDD international conference on Knowledge discovery and data mining, KDD*, pp. 947–956, 2009.

Vinicius Zambaldi, David Raposo, Adam Santoro, Victor Bapst, Yujia Li, Igor Babuschkin, Karl Tuyls, David Reichert, Timothy Lillicrap, Edward Lockhart, et al. Deep reinforcement learning with relational inductive biases. In *7th International Conference on Learning Representations, ICLR*, 2019.

Zhibin Zhao, Tianfu Li, Jingyao Wu, Chuang Sun, Shibin Wang, Ruqiang Yan, and Xuefeng Chen. Deep learning algorithms for rotating machinery intelligent diagnosis: An open source benchmark study. *arXiv preprint arXiv:2003.03315*, 2020.

Bolei Zhou, Alex Andonian, Aude Oliva, and Antonio Torralba. Temporal relational reasoning in videos. In *Proceedings of the European Conference on Computer Vision (ECCV)*, pp. 803–818, 2018.

## A    Pseudo-code of SelfTime

The overview of training process for SelfTime is summarized in Algorithm 2.

## B    Architecture Diagram

SelfTime consists of a backbone encoder, a inter-sample relation reasoning head, and a intra-temporal relation reasoning head. The detail architectural diagrams of SelfTime are shown in Table 4.

## C    Data Augmentation

In this section, we list the configuration details of augmentation used in the experiment:

---

Algorithm 2: SelfTime

---

**Require:**
   Time series set $\mathcal{T} = \{\boldsymbol{t}_n\}_{n=1}^N$.
   $f_\theta$: Encoder backbone.
   $r_\mu$: Inter-sample relation reasoning head.
   $r_\varphi$: Intra-temporal relation reasoning head.
**Ensure:**
   $f_\theta$: An updated encoder backbone.
1: **for** $\boldsymbol{t}_m, \boldsymbol{t}_n \in \mathcal{T}$ **do**
2:    Generate two augmentation sets $\mathcal{A}(\boldsymbol{t}_m)$ and $\mathcal{A}(\boldsymbol{t}_n)$
3:    Sample positive relation pair $(\boldsymbol{t}_m^{(i)}, \boldsymbol{t}_m^{(j)})$ and negative
        relation pair $(\boldsymbol{t}_m^{(i)}, \boldsymbol{t}_n^{(j)})$ from $\mathcal{A}(\boldsymbol{t}_m)$ and $\mathcal{A}(\boldsymbol{t}_n)$
4:    $\boldsymbol{z}_m^{(i)} = f_\theta(\boldsymbol{t}_m^{(i)})$                                                       ▷ Sample representation
5:    $\boldsymbol{z}_m^{(j)} = f_\theta(\boldsymbol{t}_m^{(j)})$                                                       ▷ Sample representation
6:    $\boldsymbol{z}_n^{(j)} = f_\theta(\boldsymbol{t}_n^{(j)})$                                                       ▷ Sample representation
7:    $h_{2m-1}^{(i,j)} = r_\mu([\boldsymbol{z}_m^{(i)}, \boldsymbol{z}_m^{(j)}])$                          ▷ Reasoning score of positive relation
8:    $h_{2m}^{(i,j)} = r_\mu([\boldsymbol{z}_m^{(i)}, \boldsymbol{z}_n^{(j)}])$                             ▷ Reasoning score of negative relation
9:    Sample time piece relation pair $(\boldsymbol{p}_{n,u}, \boldsymbol{p}_{n,v})$ by Algorithm 1
10:   $\boldsymbol{z}_{n,u} = f_\theta(\boldsymbol{p}_{n,u})$                                                       ▷ Time piece representation
11:   $\boldsymbol{z}_{n,v} = f_\theta(\boldsymbol{p}_{n,v})$                                                       ▷ Time piece representation
12:   $h_n^{(u,v)} = r_\varphi([\boldsymbol{z}_{n,u}, \boldsymbol{z}_{n,v}])$                            ▷ Reasoning score of intra-temporal relation
13: **end for**
14: $\mathcal{L}_{inter} = -\sum_{n=1}^{2N} \sum_{i=1}^K \sum_{j=1}^K (y_n^{(i,j)} \cdot \log(h_n^{(i,j)})$
        $+ (1 - y_n^{(i,j)}) \cdot \log(1 - h_n^{(i,j)}))$                      ▷ Inter-sample relation reasoning loss
15: $\mathcal{L}_{intra} = -\sum_{n=1}^N y_n^{(u,v)} \cdot \log \frac{\exp(h_n^{(u,v)})}{\sum_{c=1}^C \exp(h_n^{(u,v)})}$        ▷ Intra-temporal relation reasoning loss
16: Update $f_\theta$, $r_\mu$, and $r_\varphi$ by minimizing
        $\mathcal{L} = \mathcal{L}_{inter} + \mathcal{L}_{intra}$
17: **return** encoder backbone $f_\theta$, throw away $r_\mu$, and $r_\varphi$

---

**Jittering**: We add the gaussian noise to the original time series, where noise is sampled from a Gaussian distribution $\mathcal{N}(0, 0.2)$.

**Scaling**: We multiply the original time series with a random scalar sampled from a Gaussian distribution $\mathcal{N}(0, 0.4)$.

**Cutout**: We replace a random 10% part of the original time series with zeros and remain the other parts unchanged.

**Magnitude Warping**: We multiply a warping amount determined by a cubic spline line with 4 knots on the original time series at random locations and magnitudes. The peaks or valleys of the knots are set as $\mu = 1$ and $\sigma = 0.3$ (Um et al., 2017).

**Time Warping**: We set the warping path according to a smooth cubic spline-based curve with 8 knots, where the random magnitudes is $\mu = 1$ and a $\sigma = 0.2$ for each knot (Um et al., 2017).

**Window Slicing**: We randomly crop 80% of the original time series and interpolate the cropped time series back to the original length (Le Guennec et al., 2016).

**Window Warping**: We randomly select a time window that is 30% of the original time series length, and then warp the time dimension by 0.5 times or 2 times (Le Guennec et al., 2016).

## D  BASELINES

*Triplet Loss*[5] (Franceschi et al., 2019) We download the authors' official source code and use the same backbone as SelfTime, and set the number of negative samples as 10. We use Adam optimizer with learning rate 0.001 according to grid search and batch size 128 as same with SelfTime.

*Deep InfoMax*[6] (Hjelm et al., 2019) We download the authors' official source code and use the same backbone as SelfTime, and set the parameter $\alpha = 0.5, \beta = 1.0, \gamma = 0.1$ through grid search. We use Adam optimizer with learning rate 0.0001 according grid search and batch size 128 as same with SelfTime.

*Forecast*[7] (Jawed et al., 2020) Different from the original multi-task model proposed by authors, we throw away the supervised classification branch and use only the proposed forecasting branch to learn the representation in a fully self-supervised manner. We use Adam optimizer with learning rate 0.01 according to grid search and batch size 128 as same as SelfTime.

*Transformation*[8] (Sarkar & Etemad, 2020) We refer to the authors' official source code and reimplement it in PyTorch by using the same backbone and two-layer projection head as same with SelfTime. We use Adam optimizer with learning rate 0.001 according to grid search and batch size 128 as same with SelfTime.

*SimCLR*[9] (Chen et al., 2020) We download the authors' official source code by using the same backbone and two-layer projection head as same with SelfTime. We use Adam optimizer with learning rate 0.5 according grid search and batch size 128 as same as SelfTime.

*Relation*[10] (Patacchiola & Storkey, 2020) We download the authors' official source code by using the same backbone and relation module as same with SelfTime. For augmentation, we set $K = 16$, and use Adam optimizer with learning rate 0.5 according to grid search and batch size 128 as same with SelfTime.

## E  PARAMETER SENSITIVITY

Figure 7 shows the impact of different augmentation number $K$ on all datasets. It's obvious that more augmentations result in better performance, which demonstrates that introducing more reference

---

[5]https://github.com/White-Link/UnsupervisedScalableRepresentationLearningTimeSeries

[6]https://github.com/rdevon/DIM

[7]https://github.com/super-shayan/semi-super-ts-clf

[8]https://code.engineering.queensu.ca/17ps21/SSL-ECG

[9]https://github.com/google-research/simclr

[10]https://github.com/mpatacchiola/self-supervised-relational-reasoning

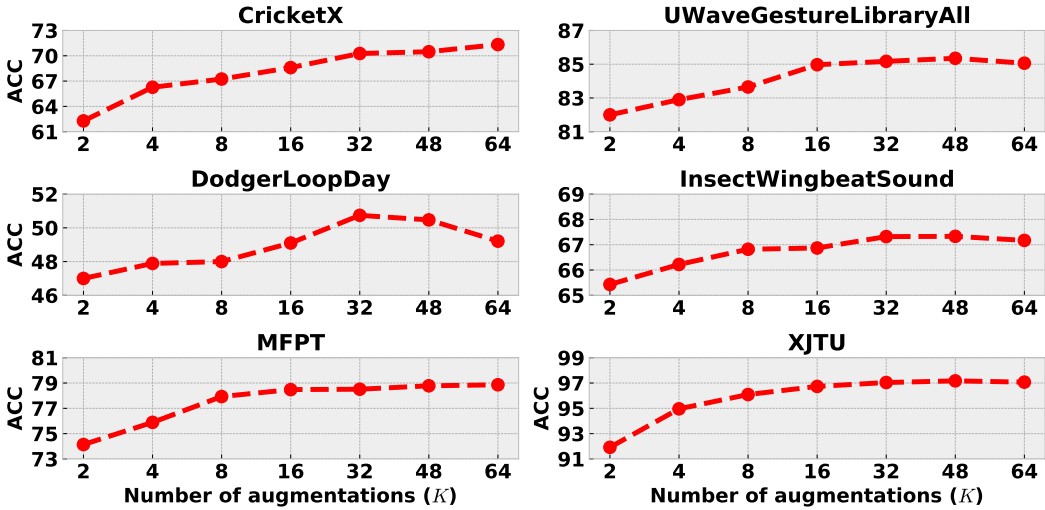

Figure 7: Impact of augmentation number $K$.

samples (including positive samples and negative samples) for the anchor sample raises the power of relational reasoning.

Figure 8 shows the impact of different temporal relation class numbers and piece sizes on other five datasets: UWaveGestureLibraryAll, DodgerLoopDay, InsectWingbeatSound, MFPT, and XJTU, where the blue bar indicates class reasoning accuracy on training data (Class ACC) and the brown line indicates the linear evaluation accuracy on test data (Linear ACC). We find an interesting phenomenon is that both small values of class number $C$ or piece size $L/T$, and big values $C$ or $L/T$, result in worse performance. One possible reason behind this is that the increase of class number drops the Class ACC and makes the relation reasoning task too simple (with high Class ACC) or too difficult (with low Class ACC) and prevents the network from learning useful semantic representation. Therefore, an appropriate pretext task designing is crucial for the self-supervised time series representation learning. In the experiment, to select a moderately difficult pretext task for different datasets, we set {class number ($C$), piece size ($L/T$)} as {4, 0.2} for UWaveGestureLibraryAll, {5, 0.35} for DodgerLoopDay, {6, 0.4} for InsectWingbeatSound, {4, 0.2} for MFPT, and {4, 0.2} for XJTU.

# F ABLATION STUTY

In this section, we additionally explore the effectiveness of different relation reasoning modules including inter-sample relation reasoning, intra-temporal relation reasoning, and their combination (SelfTime) on other five datasets including UWaveGestureLibraryAll, DodgerLoopDay, InsectWingbeatSound, MFPT, and XJTU. Specifically, in the experiment, we systematically investigate the different data augmentations on the impact of linear evaluation for different modules. Here, we consider several common augmentations including magnitude domain based transformations such as jittering (Jit.), cutout (Cut.), scaling (Sca.), magnitude warping (M.W.), and time domain based transformations such as time warping (T.W.), window slicing (W.S.), window warping (W.W.). Figure 9 and Figure 10 show linear evaluation results on five datasets under individual and composition of transformations for inter-sample relation reasoning, intra-temporal relation reasoning, and their combination (SelfTime). As similar to the observations from CricketX, firstly, we observe that the composition of different data augmentations is crucial for learning useful representations. For example, inter-sample relation reasoning is more sensitive to the augmentations, and performs worse under Cut., Sca., and M.W. augmentations, while intra-temporal relation reasoning is less sensitive to the manner of augmentations on all datasets. Secondly, by combining both the inter-sample and intra-temporal relation reasoning, the proposed SelfTime achieves better performance, which demonstrates the effectiveness of considering different levels of relation for time series representation learning. Thirdly, overall, we find that the composition from a magnitude-

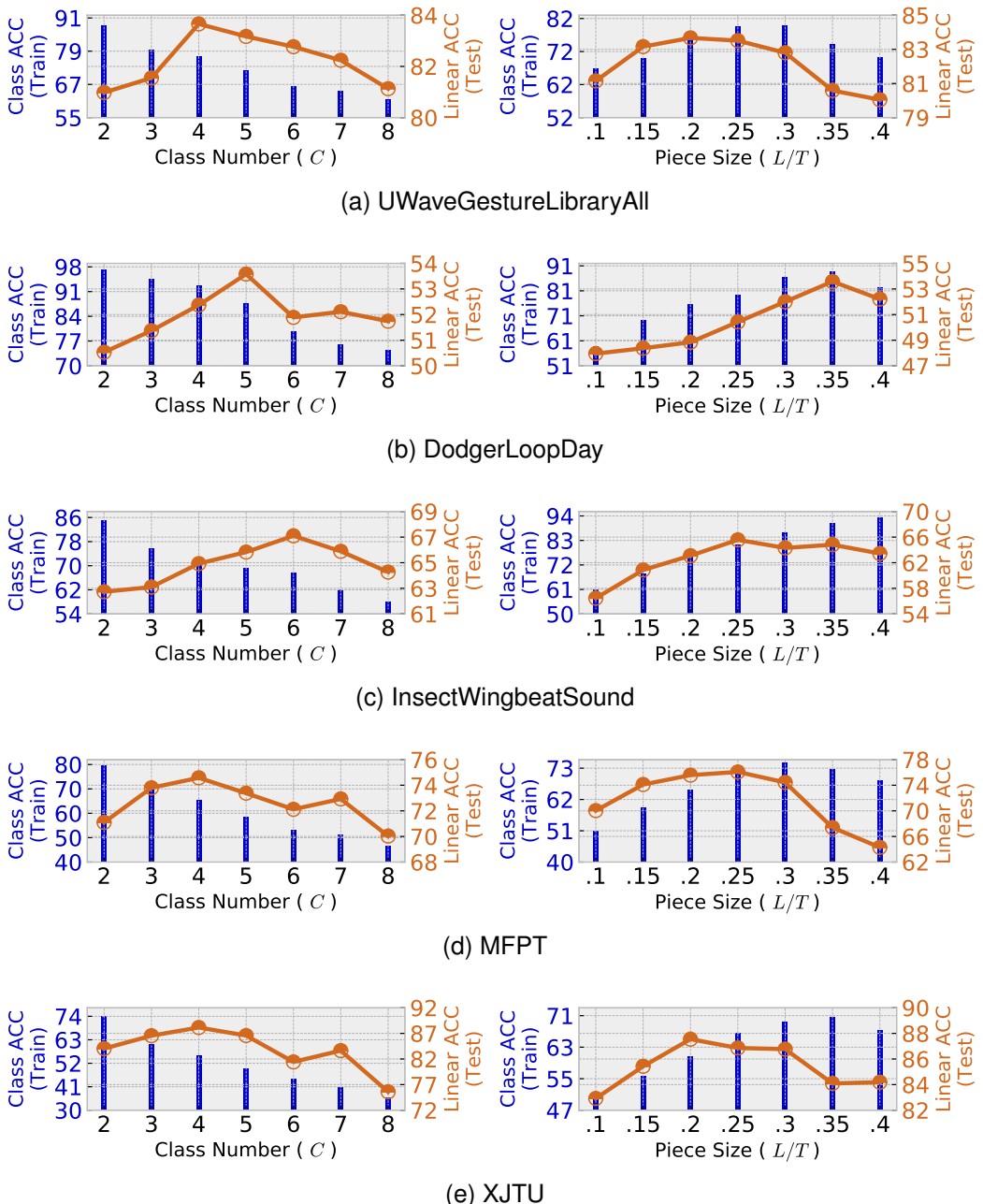

(a) UWaveGestureLibraryAll

(b) DodgerLoopDay

(c) InsectWingbeatSound

(d) MFPT

(e) XJTU

Figure 8: Impact of different temporal relation class numbers and piece sizes on other five datasets.

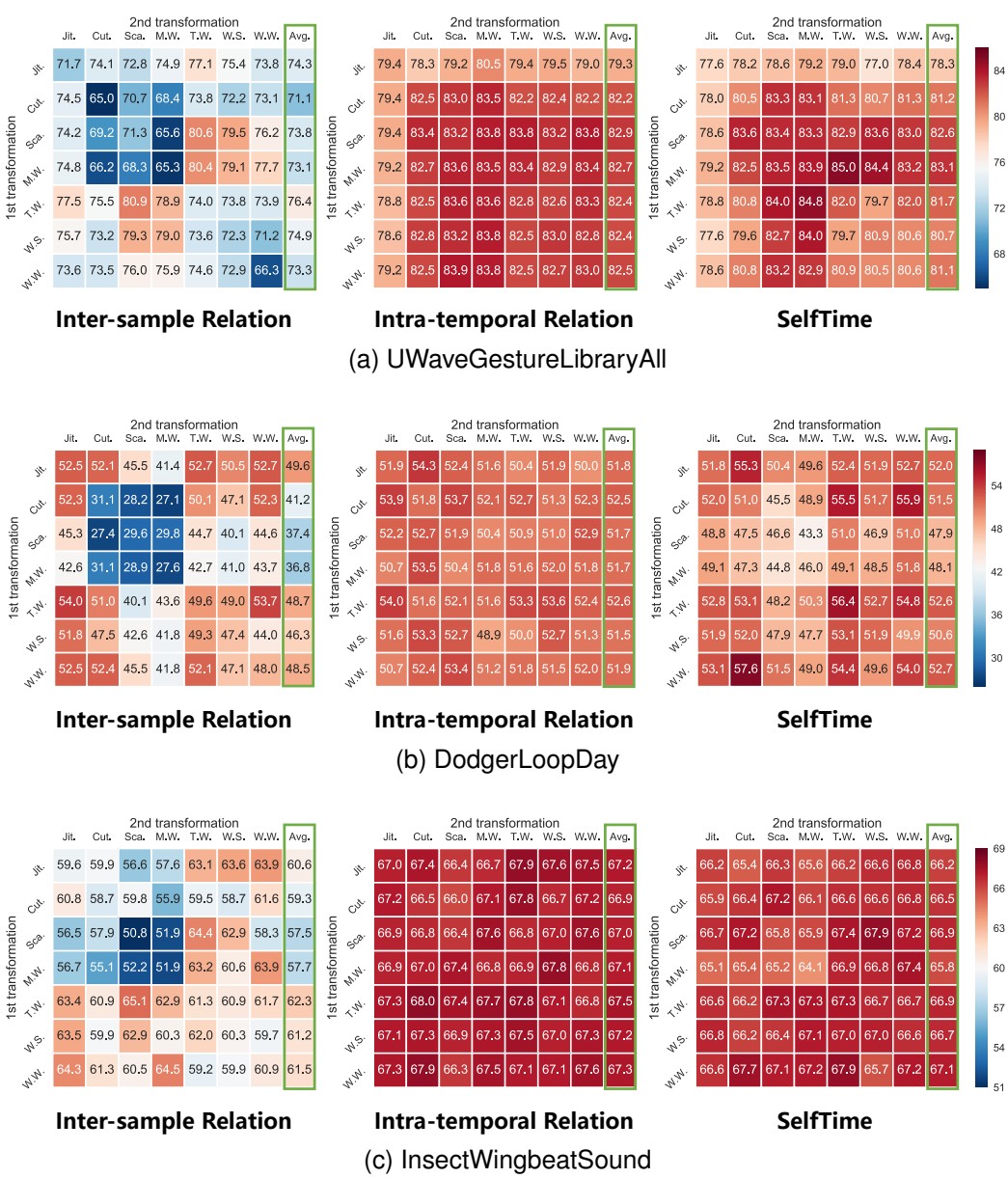

Figure 9: Linear evaluation on UWaveGestureLibraryAll, DodgerLoopDay, and InsectWingbeat-Sound datasets under individual or composition of data augmentations. For all columns but the last, diagonal entries correspond to single transformation, and off-diagonals correspond to composition of two transformations (applied sequentially). The last column reflects the average over the row.

based transformation (e.g. scaling, magnitude warping) and a time-based transformation (e.g. time warping, window slicing) facilitates the model to learn more useful representations. Therefore, in this paper, we select the composition of magnitude warping and time warping augmentations for all datasets, although other compositions might result in better performance.

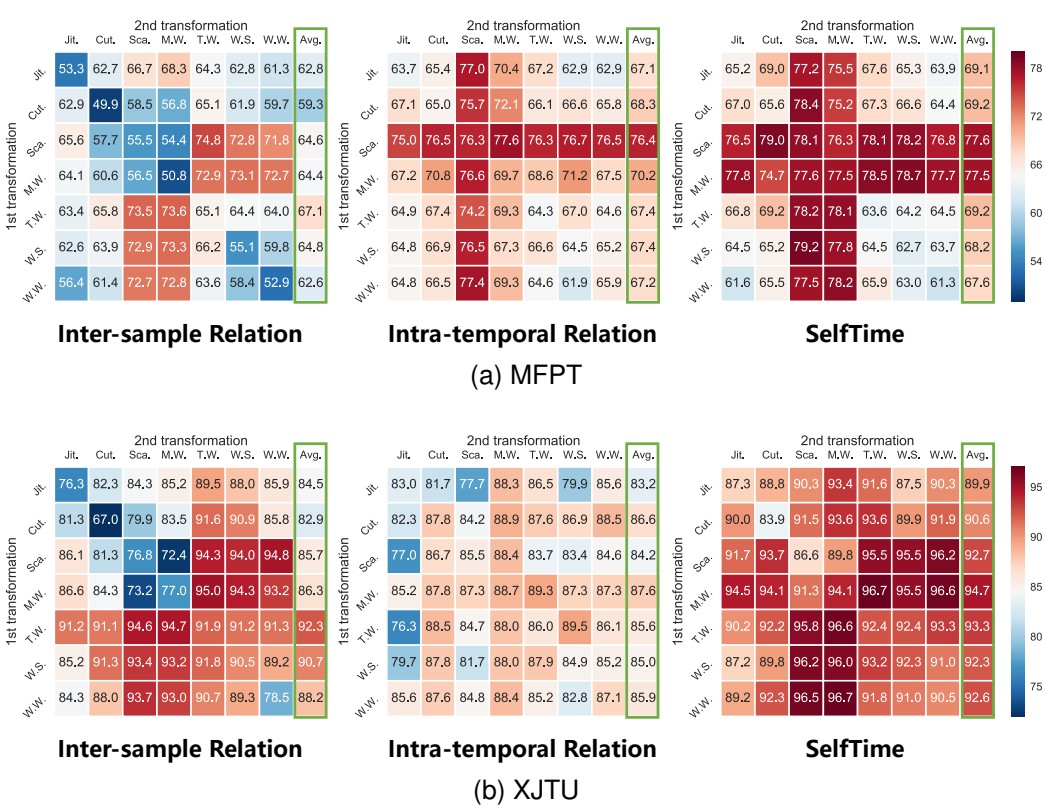

Figure 10: Linear evaluation on MFPT and XJTU datasets under individual or composition of data augmentations. For all columns but the last, diagonal entries correspond to single transformation, and off-diagonals correspond to composition of two transformations (applied sequentially). The last column reflects the average over the row.

