# OpenReview forum: "Self-Supervised Time Series Representation Learning by Inter-Intra Relational Reasoning"
_ICLR.cc/2021/Conference — Reject_

### Official Review · AnonReviewer2 · 2020-10-28

**Rating:** 5
**Confidence:** 4

**Review:**

===========Update after rebuttal==================

I will remain my score of weak rejection.


=============================================================

This paper presents to model both inter-sample and intra-temporal relations. The idea is naive but is easy to follow and reasonable. Experimental results support the effectiveness of the method.

Strengths: The idea is reasonable and the application is important.

Weakness:
1. The notations are messy. The author should consider clean up some unused notations for better readability. Especially for section 3.1, I spend quite a few time thoroughly digest the notations. The explanation in section 3.2 is even scarier. I can easily understand the concept from the intro and Figure 1, yet I spend much more time reading sections 3.1 and 3.2.
2. The presentation of the experimental section is also hard to follow. Specifically, what is the message conveyed in Figure 5? I would suggest the author highlight the region that we can pay attention to. Nonetheless, why only considering the composition of two augmentations? Instead of providing a matrix that considers all the combinations between two different augmentations, why not providing a table summarizing the sets of different combinations of augmentations? The sets of combinations can be the sets that the author likes us to focus on.
3. An important paper [1] is missing. The author should discuss this missing reference.

Overall, I am leaning positive toward this paper, yet I feel the presentation can be greatly improved. I am giving the score 5 for now, and I may update the score after the rebuttal.

[1] Temporal Relational Reasoning in Videos, Zhou et al., ECCV 2018.

---

> ### Author Response · Authors · 2020-11-24
> **Response #1**
>
>
> **On presentation of method parts:**
> We have revised Section 3 according to the review's suggestion. In the revised version, we clean up the unused notations and rewrite some sentences to make the description more clearer. More details are shown in the marked parts in the revised version.
>
> **What is the message conveyed in Figure 5?**
>
> Figure 5 shows the experimental results about the impact of different data augmentations and relation modules.
>
>
> Firstly, to evaluate the impact of different data augmentations, as similar to SimCLR [1], we conduct linear evaluation under individual or composition of data augmentations including several common time series augmentations such as magnitude domain based transformations such as jittering (Jit.), cutout (Cut.), scaling (Sca.), magnitude warping (M.W.), and time domain based transformations such as time warping (T.W.), window slicing (W.S.), window warping (W.W.). We observe that the composition of different data augmentations is crucial for learning useful representations. For example, inter-sample relation reasoning is more sensitive to the augmentations, and performs worse under Cut., Sca., and M.W. augmentations, while intra-temporal relation reasoning is less sensitive to the manner of augmentations, although it performs better under the time domain based transformation.
>
>
> Secondly, to evaluate the effectiveness of different relation modules for ablation study, we compare the linear evaluation results on different modules including Inter-sample Relation, Intra-temporal Relation and their combination (SelfTime). We find that, by combining both the inter-sample and intra-temporal relation reasoning, the proposed SelfTime achieves better performance, which demonstrates the effectiveness of considering different levels of relation for time series representation learning.
>
> Similar experimental conclusions also hold on for other datasets. More experimental results on the other five datasets for evaluation of the impact of different relation modules and data augmentations are shown in Appendix F.
>
>
> **Why only considering the composition of two augmentations?**
>
> As similar to SimCLR [1], to better understand the effects of individual data augmentations and the importance of augmentation composition, we conduct linear evaluation when applying augmentations individually or in pairs. Previous studies show that those contrastive learning [1] or inter-sample relation reasoning [2] based methods are sensitive to the augmentation strategies on image data, therefore, it's also necessary to evaluate it on time series data. Here, we use several common time series augmentations such as magnitude domain based transformations such as jittering (Jit.), cutout (Cut.), scaling (Sca.), magnitude warping (M.W.), and time domain based transformations such as time warping (T.W.), window slicing (W.S.), window warping (W.W.). We observe that the composition of different data augmentations is crucial for learning useful time series representations. For example, inter-sample relation reasoning is more sensitive to the augmentations, and performs worse under Cut., Sca., and M.W. augmentations, while intra-temporal relation reasoning is less sensitive to the manner of augmentations, although it performs better under the time domain based transformation.
>
> In summary, our main purpose is to evaluate that the impact of different augmentation composition is important, not the purpose of the best composition for evaluation because there could be plenty of choices. From the experimental results, we can obviously find that inter-sample relation module is more sensitive to augmentations while intra-temporal relation module does not, while intra-temporal module performs better than the inter-sample module in most cases, which provides our more insights into pretext task designing on time series data. Therefore, we believe that conducting augmentations in pairs is enough to support our purpose.
>
>
>
>     [1] Chen, Ting, et al. "A simple framework for contrastive learning of visual representations." ICML'20 (2020).
>     [2] Self-supervised relational reasoning for representation learning. Patacchiola and Storkey. NeurIPS'20

---

> ### Author Response · Authors · 2020-11-24
> **Response #2**
>
>
> **An important paper [1] is missing. The author should discuss this missing reference.**
>
> We have added this missing work [1] in the Introduction and Related Work section.
>
> Temporal Relation Network (TRN) is proposed in [1] to learn the temporal dependencies between video frames at multiple time scales, by replacing the average pooling operation with an interpretable relational module in order to better capture information along the temporal dimension. However, the main difference between TRN and SelfTime is that, TRN is a supervised method for video action recognition, while SelfTime is a self-supervised method for time series representation learning. Different form TRN, which randomly sample different ordered frame temporal relations during feature extraction under the supervised video action classification loss, SelfTime randomly sample the intra-sample temporal relation based on the time pieces and supervised the training of network in a self-supervised manner. Overall, SelfTime is a far different self-supervised representation framework compared with the supervised TRN.
>
>
> [1] Temporal Relational Reasoning in Videos, Zhou et al., ECCV 2018.

---

### Official Review · AnonReviewer4 · 2020-10-28
**Official Blind Review**

**Rating:** 5
**Confidence:** 4

**Review:**

The paper proposes an approach for self-supervised time series representation learning by using inter-sample and intra-temporal relational reasoning. The paper builds upon the existing ideas on relational reasoning [3] and self-supervised learning to train models from unlabeled data.
Self-supervised learning for time series is still under-explored and this paper attempts to bridge this gap in time series literature.

The key novelty of the paper as claimed by the authors is in the design of inter-sample and intra-temporal tasks and loss functions to learn a feature extractor from unlabeled data.
This idea of designing pretext tasks using global-sample structure and local-temporal structure and adapting it for time series is interesting. However, the inter-sample loss function seems to be a straightforward application of ideas in self-supervised learning literature that rely on various augmentations to create positive and negative pairs of samples. The intra-temporal task is defined where the distance between subsequences (referred to as time pieces in the paper) of a time series is used to create a classification task. This is a potentially novel (albeit incremental) aspect of the approach but the empirical evaluations (as detailed below) fail to highlight the impact of the same on performance clearly.
Furthermore, the idea of using subsequences to define self-supervised (intra-temporal) tasks for time series has been explored earlier, e.g. in [1,2]. The authors seem to be unaware of papers like [1,2], and Introduction and Related Work sections suggest that self-supervised learning for time series has not been attempted earlier.

Apart from lack of clarity on novelty and contribution of the work, I have following concerns regarding empirical evaluation:
1. The ablation studies show the effect of number of classes and the length of subsequences (pieces) on linear evaluation accuracy (Linear ACC) for the downstream classification tasks. However, the sensitivity of results to the hyperparameters of intra-temporal task is reported only on CricketX. Do the same results hold on other datasets? Also, the results on downstream task seem to be very sensitive to the choice of parameters C and L/T. The authors state that "In the experiment, to select a moderately difficult pretext task for different datasets, we set {class number (C), piece size (L/T )} as {3, 0.2} for CricketX, {4, 0.2} for UWaveGestureLibraryAll, {5, 0.35} for DodgerLoopDay, {6, 0.4} for InsectWingbeatSound, {4, 0.2} for MFPT, and {4, 0.2} for XJTU" - It is not clear how these choices for hyperparameters are arrived at, or what "we set"  and "moderately difficult pretext task" mean. Also, the ablation study is more of a sensitivity analysis of the choice of values for C and L/T. It is not clear what happens if intra-temporal relation reasoning task or the inter-sample relation reasoning task is removed from SelfTime. Does that make SelfTime same as one of the other baselines, e.g. "Relation"? Since most of the baselines used are adaptations from image domain, it is difficult to gauge where the proposed approach stands w.r.t. time-series specific methods like [1,2].
2. For CricketX, DLD, and XJTU datasets, SelfTime performs significantly better than the supervised learning model (Table 2). How does one explain this observation as supervised methods would typically perform better or at least as good as the self-supervised methods?
3. The authors observe that "we find that the composition from a magnitude-based transformation (e.g. scaling, magnitude warping) and a time-based transformation (e.g. time warping, window slicing) facilitates the model to learn more useful representations." As per the description, it seems that this observation is based on analysis of just one dataset (CricketX) and on one algorithm (SelfTime). I think a more thorough description and evaluation across datasets and baselines could be useful.
4. In the ablation results, it would help to see results across all datasets when using only inter-sample or intra-temporal losses. Also, the results in Fig. 5 and Fig. 6 are for only one dataset, and that too different ones: CricketX for Fig. 5 and UGLA for Fig. 6. Do the observations from Fig. 5 and 6 hold across datasets? Similarly, what motivates the particular pairing of source-->target for the Domain Transfer Evaluation? Given six datasets, several other combinations are possible - does this observation hold across all such combinations?
5. The authors loosely mention that "more augmentation results in better performance" on the basis of references from existing literature. Though this might hold in other domains, an evaluation of the same for this setting would be useful to claim the same in the given context.

Given the above, a more thorough and consistent description of the contribution in light of recent related work and empirical evaluation is needed. Given works like [1,2,3], the contribution of the paper is limited to defining the intra-temporal task, but the same has not been evaluated carefully enough.

The write-up can be improved at several places, e.g.:
Eqns. 1 and 2 need a minus sign to be called as losses?
Is i=j valid for the positive inter-sample relation pair? If not, Eqn. 1 might need an update.
fire --> fair?
$z_n$ is bold at a few places and not at other
limited amount training samples
Or for video data
there is not enough feature
we generate from its transformation counterpart and another individual sample as the positive and negative samples respectively
In the experiment, we achieve new state-of-the-art results, and outperforms existing methods by a significant margin on multiple real-world time series datasets for classification task
a intra-temporal
Firstly, takes the original time series signals and their sampled time pieces as the inputs
show that both two kinds of relation reasoning
will drop the evaluation performance
t-SNE visualization of the learnt feature.

References:
[1] Unsupervised scalable representation learning for multivariate time series. Franceschi et. al. NeurIPS'19
[2] Multi-task Self-Supervised Learning for Human Activity Detection. Saeed et. al. PACM on Interactive, Mobile, Wearable and Ubiquitous Technologies 2019.
[3] Self-supervised relational reasoning for representation learning. Patacchiola and Storkey. NeurIPS'20

** updating the score to 5 **

---

> ### Author Response · Authors · 2020-11-24
> **Response #1**
>
> Thanks for the comments and questions! Please find below our answers.
>
>
> **On difference between relation reasoning loss and contrastive loss:**
>
> Different from the pair-wise distance based contrastive loss [1] and cross-entropy based sample classification loss [2,3] in the traditional self-supervised models, inter-sample relation reasoning [4] leverages a sample-relation based loss for binary positive-negative relation classification. Although inter-sample loss is a binary cross-entropy based loss, its optimized object is sample relation representation, not the sample representation as in those traditional methods like [2,3]. Also, we leverage binary cross-entropy based loss for more efficient binary relation learning compared with the pair-wise distance based contrastive loss [1].
>
> The pairwise-distance based contrastive loss relies on a large number of negatives, which are difficult to obtain in mini-batch stochastic optimization, and SimCLR requires specialized optimizers (e.g., cosine annealing) to stabilize the training at scale, however, the cross-entropy do not need the convoluted sample-mining heuristics, resulting in a more easy optimization process. Although cross-entropy does not explicitly involve pairwise distances, as demonstrated in [5], it's an upper bound on a new pairwise loss, which has a structure similar to various pairwise losses: it minimizes intra-class distances while maximizing inter-class distances. As a result, minimizing the cross-entropy can be seen as an approximate bound-optimization (or Majorize-Minimize) algorithm for minimizing this pairwise loss. Please refer [5] for more details about the relationship between cross-entropy loss and pairwise-distance loss.
>
>
>     [1] Chen, Ting, et al. "A simple framework for contrastive learning of visual representations." ICML'20 (2020).
>     [2] Multi-task Self-Supervised Learning for Human Activity Detection. Saeed et. al. PACM on Interactive, Mobile, Wearable and Ubiquitous Technologies 2019.
>     [3] Sarkar, Pritam, and Ali Etemad. "Self-supervised learning for ecg-based emotion recognition." ICASSP 2020-2020 IEEE International Conference on Acoustics, Speech and Signal Processing (ICASSP). IEEE, 2020.
>     [4]  Patacchiola and Storkey. “Self-supervised relational reasoning for representation learning.” NeurIPS'20
>     [5] Boudiaf, Malik, et al. "A unifying mutual information view of metric learning: cross-entropy vs. pairwise losses." European Conference on Computer Vision. Springer, Cham, 2020.
>
>
> **The authors seem to be unaware of papers like [1,2]}, and Introduction and Related Work sections suggest that self-supervised learning for time series has not been attempted earlier.**
>
> We have added the missing related works [1, 2] and other related self-supervised time series representation learning works in the Introduction and Related Work section. More details are shown in the marked parts of the revised version.
>
> In [1], the paper follows word2vec’s intuition by assuming that the context of a subsequence of a time series should probably be close to the one of the same time series, and distant from the one of randomly chosen time series, since they are probably unrelated to the original time series’s context. Then, they use triplet loss to push pairs of (subsequence, context) and (subsequence, random context) to be linearly separable for unsupervised time series representation learning. Different from [1], in our intra-temporal module, we attempt to capture the temporal pattern by reasoning the multi-level relation among time pieces sampled from the same time series, while [1] focuses on the binary relation among time pieces sampled from the different time series.
>
> In [2], the paper proposes a Transformation Prediction Network, where a temporal convolutional neural network is trained jointly on different signal transformations to solve a problem of transformation recognition. In [2], there is no explicitly modeling of temporal pattern, while in our intra-temporal module, we attempt to capture the temporal pattern by reasoning the temporal relation among time pieces sampled from each time series.
>
>     [1] Unsupervised scalable representation learning for multivariate time series. Franceschi et. al. NeurIPS'19
>     [2] Multi-task Self-Supervised Learning for Human Activity Detection. Saeed et. al. PACM on Interactive, Mobile, Wearable and Ubiquitous Technologies 2019.

---

> > ### Author Response · Authors · 2020-11-24
> > **Response #2**
> >
> > **On ablation study of different relation reasoning tasks:**
> >
> > In the ablation study section, we firstly evaluate parameter sensitivity analysis to demonstrate the impact of two important hyperparameters C and L/T. Then, we evaluate the impact/effectiveness of different pretext tasks by removing intra-temporal relation reasoning task (inter-sample relation) or the inter-sample relation reasoning task (intra-temporal relation) from SelfTime, and the experimental results on CricketX are shown in Figure 5.
> >
> > We have also added ablation study experiments on all other datasets in the Appendix F.  As similar to the observations from CricketX, firstly, we observe that the composition of different data augmentations is crucial for learning useful representations. For example, inter-sample relation reasoning is more sensitive to the augmentations, and performs worse under Cut., Sca., and M.W. augmentations, while intra-temporal relation reasoning is less sensitive to the manner of augmentations on all datasets. Secondly, by combining both the inter-sample and intra-temporal relation reasoning, the proposed SelfTime achieves better performance, which demonstrates the effectiveness of considering different levels of relation for time series representation learning. Thirdly, overall, we find that the composition from a magnitude-based transformation (e.g. scaling, magnitude warping) and a time-based transformation (e.g. time warping, window slicing) facilitates the model to learn more useful representations.
> >
> > **On performance comparison with time series based baselines:**
> >
> > We have added two time series based baselines [1, 3] in the experiment, where [1] is an unsupervised time series representation learning model that uses triplet loss to push a subsequence of time series close to its context and distant from a randomly chosen time series. [3] is a self-supervised model that is similar to [2] by designing transformation recognition of different time series transformations as the pretext task. Experimental results are shown in Table 2 and Table 3, which demonstrate that SelfTime consistently outperforms all baselines because of its effective modeling of different levels of relation among time series.
> >
> >     [1] Unsupervised scalable representation learning for multivariate time series. Franceschi et. al. NeurIPS'19
> >     [2] Multi-task Self-Supervised Learning for Human Activity Detection. Saeed et. al. PACM on Interactive, Mobile, Wearable and Ubiquitous Technologies 2019.
> >     [3] Sarkar, Pritam, and Ali Etemad. "Self-supervised learning for ecg-based emotion recognition." ICASSP 2020-2020 IEEE International Conference on Acoustics, Speech and Signal Processing (ICASSP). IEEE, 2020.
> >
> >
> > **How does one explain this observation as supervised methods would typically perform better or at least as good as the self-supervised methods?**
> >
> > In the computer vision domain, supervised models typically perform better than or as similar as self-supervised models. However, in other domains such as audio [1] or time series [2, 3], the-state-of-the-art self-supervised models perform better than supervised methods, and even by a significant margin on some specific datasets.
> >
> > In our work, SelfTime outperforms the supervised baseline by a large margin on Cricket and DLD datasets, and by a small margin on IWS and XJTU datasets. The possible reasons for this phenomenon might be two folds: 1. The two datasets Cricket and DLD are small, therefore, the supervised baseline would like to be overfitting on training data because of the limited training data, which results in poor performance. 2. Different from the supervised method, our proposed SelfTime tries to capture the underlying temporal structure of time series for more robust feature learning, which results in better performance.
> >
> >
> >     [1] Shukla, Abhinav, Stavros Petridis, and Maja Pantic. "Learning Speech Representations from Raw Audio by Joint Audiovisual Self-Supervision." arXiv preprint arXiv:2007.04134 (2020).
> >     [2] Sarkar, P., \& Etemad, A. (2020). Self-supervised ecg representation learning for emotion recognition. arXiv preprint arXiv:2002.03898.
> >     [3] Saeed, Aaqib, Tanir Ozcelebi, and Johan Lukkien. "Multi-task self-supervised learning for human activity detection." Proceedings of the ACM on Interactive, Mobile, Wearable and Ubiquitous Technologies 3.2 (2019): 1-30.
> >
> >
> > **What motivates the particular pairing of source-->target for the Domain Transfer Evaluation?**
> >
> > In the experiment, we select two datasets from the same category as source and target respectively. The domain transferring experiments are conducted between datasets owning the same category. We have added the other three groups of experimental in Table 3.

---

> > > ### Author Response · Authors · 2020-11-24
> > > **Response #3**
> > >
> > >
> > > **On evaluation of the statement that more augmentation results in better performance**
> > >
> > > We have added new experiments to evaluate the impact of augmentation numbers $K$ in Figure 7 of Appendix E. From the experiment, it's obvious that more augmentations result in better performance, which demonstrates that introducing more reference samples (including positive samples and negative samples) for the anchor sample raises the power of relational reasoning.
> > >
> > >
> > > **The write-up can be improved at several places.**
> > >
> > > We have corrected the typo errors and rewritten some parts according to the suggestion of reviewers. More details are shown in the marked parts in the revised version.

---

> > > > ### Comment · AnonReviewer4 · 2020-11-24
> > > > **Thanks for detailed response. Few concerns remain.**
> > > >
> > > > Thanks for the significant updates to the paper.
> > > >
> > > > I have a few concerns:
> > > > 1. The concern 1 in the review still remains unaddressed: "to select a moderately difficult pretext task for different datasets" is used to choose the class number and piece size, especially as the results in Fig. 8 suggest that the training and test performances are not always correlated. It is not clear how this can be done in practice, and seems to be a major limitation.
> > > > 2. The empirical comparison with [1] and [2] is appreciated. However, I am concerned about the claims of state-of-the-art in the paper, and if the comparison with [1] is fair as most of the numbers from [1] seem to be underreported, or being compared in a different setting - the accuracy numbers for various datasets (UWGLA-94.1%, IWS-62.3%, and CricketX-77.7%) as reported in Table S1 of the Supplementary Material of [1] are significantly higher than those reported in this paper in Table 2. A clarification on this will be useful.
> > > > 3. Another concern is that while [1] evaluates on around 85 datasets from UCR Archive, this work compares on just 4 of those datasets (and two additional datasets). It has been long argued now in the time series classification literature that if using UCR Archive for evaluation, it is better to report results on a significant number (usually all) of datasets as it is always possible to find 3-4 datasets where a particular approach would do better than all or most of the other approaches, unless it is known beforehand on the nature/kind/characteristics of datasets where the proposed approach will do better than others. Given this and the concern-2 above, it is difficult to gauge the claim on state-of-the-art in this work.
> > > > 4. There are a quite a few new grammatical errors and typos.
> > > >
> > > > Given the considerable revisions which address many of the earlier concerns and the above unaddressed/new concerns, revising the score from 3 to 5 - I feel the contributions are interesting, important, and in the right direction, but are incremental given the prior art, and need a more thorough evaluation.
> > > >
> > > > Reference:
> > > > [1] Unsupervised Scalable Representation Learning for Multivariate Time Series, NeurIPS19
> > > > https://papers.nips.cc/paper/2019/file/53c6de78244e9f528eb3e1cda69699bb-Supplemental.zip

---

> > > > > ### Author Response · Authors · 2020-11-25
> > > > > **Response**
> > > > >
> > > > > Thanks for the comments and questions! Please find below our answers.
> > > > >
> > > > > **Fig. 8 suggest that the training and test performances are not always correlated.**
> > > > >
> > > > > Although there is not a significant correlation between the training and test performances, the common characteristic among the subfigures of Fig. 8 is that either the high training ACC or the low training ACC would not result in the best test performance, and therefore, the most important information revealed by Fig. 8 is that a too simple (with high training ACC) or too difficult (with low training ACC) relation reasoning task prevents the network from learning useful semantic representation. Therefore, an appropriate pretext task designing is crucial for the self-supervised time series representation learning. In practice, for different datasets (time series data), we need to choose appropriate intra-temporal pretext task by choosing appropriate hyperparameters class number $C$ and piece size $L/T$. In summary, the purpose of Fig. 8 is to investigate the impact of the two important hyperparameters, and explain the reasons behind the experimental results.
> > > > >
> > > > >
> > > > >
> > > > > **On fair comparison with baseline [1]:**
> > > > >
> > > > > The results reported by [1] and by our work are different because the experiments are conducted under different settings.
> > > > >
> > > > > Firstly, train/test split ratios are different. [1] used the original train/test split from UCR dataset (UGLA: 0.2:0.8, IWS: 0.1:0.9, CricketX: 0.5:0.5), while in our work, we shuffle all samples and split the train-val-test ratio as 0.5:0.25:0.25 for all datasets.
> > > > >
> > > > > Secondly, the classifier is different. [1] train an SVM with radial basis function kernel on top of the learned features using the train labels of the dataset, and output the corresponding classification score on the test set. While in our work, we conduct the standard linear evaluation as same as other self-supervised works by training a linear classifier on top of the representations learned from different self-supervised models.
> > > > >
> > > > > Thirdly, the test results are measured in different manners. [1] used an SVM classifier to supervise the representation quality during training by using the train labels of the dataset, although the training loss of [1] is an unsupervised triplet loss. While in our work, we do not use any train labels during training, the supervisor signals of different models do not include train labels. (e.g., SelfTime only uses the self-generated relation labels including inter-sample relation labels and intra-temporal relation labels as the supervisor signals during training.)
> > > > >
> > > > >
> > > > >
> > > > > Fourthly, the backbone encoders are different. [1] used a complex stacked dilated causal convolutional network for feature extraction, while SelfTime uses a simple 4-layer 1D convolutional network for feature extraction.
> > > > >
> > > > >
> > > > > Therefore, it is reasonable that the results reported by [1] are far different from our reported results. In our work, as a common evaluation protocol, linear evaluation is used in the experiment by training a linear classifier on top of the representations learned from different self-supervised models to evaluate the quality of the learned embeddings. For data splitting, we set the training/validation/test split as 50\%/25\%/25\%. During the pretraining stage, we randomly split the data 5 times with different seeds, and train the backbone on them. During the linear evaluation, we train the linear classifier 10 times on each split data, and the best model on the validation dataset was used for testing. Finally, we report the classification accuracy as mean with the standard deviation across all trials. We believe our experimental settings are fair for all the compared models.
> > > > >
> > > > >
> > > > >
> > > > > **On selection of the evaluation datasets:**
> > > > >
> > > > > We select three different categories of time series (Motion, Sensor, and Device) to evaluate the generality and the effectiveness of the proposed method. For motion and sensor data, we select four typical datasets that consist of various numbers of instances, signal lengths, and the number of classes from the UCR dataset. Besides this, we also select another two commonly used real-world bearing machine datasets XJTU and MFPT for evaluation.
> > > > >
> > > > >
> > > > > **There are a quite a few new grammatical errors and typos.**
> > > > >
> > > > > Thanks for pointing it out. We carefully check out the manuscript again and correct some errors.

---

### Official Review · AnonReviewer3 · 2020-10-30
**Interesting architecture for self-supervised temporal representation learning, but the novelty is limited  compared to contrastive learning.**

**Rating:** 6
**Confidence:** 3

**Review:**

**Summary**
This paper presents a general Self-supervised Time Series representation learning framework. It explores the inter-sample relation reasoning and intra-temporal relation reasoning of time series to capture the underlying structure pattern of the unlabeled time series data.  The proposed method achieves new state-of-the-art results and outperforms existing methods by a significant margin on multiple real-world time-series datasets for the classification tasks.

**Contributions**
1. The paper is well written and easy to follow. The organization is good.
2. The architecture is well motivated. It is reasonable to use unsupervised temporal relations to learn video features.
3. The qualitative results are numerous, insightful, and convincing on multiple datasets. The authors conduct extensive experiments to demonstrate the effectiveness of the proposed method, including inter-sample relation reasoning and intra-temporal relation reasoning.

**Details**
1. The novelty of the proposed method.
The Inter-sample relation reasoning is very similar to SimCLR, which also maximizes agreement between different views of augmentation from the same sample via a contrastive loss.  Considering this, the novelty is relatively incremental.

2. Additional video recognition experiments.
The used dataset is small-scale that makes the task simple. I would have wanted to see results on large-scale classification tasks, such as video action classification.  The performance of action classification is closely related to the temporal feature modeling. So the effects on this task can make the proposed method more convincing.

**Conclusion**
overall, this paper proposes an interesting architecture for self-supervised temporal modeling. But the novelty is relatively limited compared to the recent SimCLR work. And it requires additional experiments on harder video classification task and datasets to show the effects and robustness of the proposed method.

**After rebuttal**

Thanks for the detailed response.
This paper can be seen as an interesting attempt to use self-supervised on time series data.
Although the basic idea is similar to SimCLR, It is still interesting work considering the  computation complexity and new loss function.
So I update my score to 6.

---

> ### Author Response · Authors · 2020-11-24
> **Response #1**
>
> Thanks for the comments and questions! Please find below our answers.
>
> **On novelty of the proposed method:**
>
> 1. The difference between SelfTime and SimCLR:
>     Relation reasoning is far different from the contrastive learning based methods such as SimCLR. There are two main differences between them:
>
>     1.1. Loss function: SimCLR uses the pairwise-distance based contrastive loss function as objective to guide the inter-sample relation reasoning for time series representation learning, while in SelfTime, we use the cross-entropy based loss function as objective to guide the reasoning of different levels of entity relationship (inter-sample relation and intra-temporal relation) for learning of useful time series representation.
> The pairwise-distance based contrastive loss relies on a large number of negatives, which are difficult to obtain in mini-batch stochastic optimization, and SimCLR requires specialized optimizers (e.g., cosine annealing) to stabilize the training at scale, however, the cross-entropy do not need the convoluted sample-mining heuristics, resulting in a more easy optimization process. Although cross-entropy does not explicitly involve pairwise distances, as demonstrated in [1], it's an upper bound on a new pairwise loss, which has a structure similar to various pairwise losses: it minimizes intra-class distances while maximizing inter-class distances. As a result, minimizing the cross-entropy can be seen as an approximate bound-optimization (or Majorize-Minimize) algorithm for minimizing this pairwise loss. Please refer [1] for more details about the relationship between cross-entropy loss and pairwise-distance loss.
>
>     [1] Boudiaf, Malik, et al. "A unifying mutual information view of metric learning: cross-entropy vs. pairwise losses." European Conference on Computer Vision. Springer, Cham, 2020.
>
>     1.2. Computation complexity:
> SelfTime is a more efficient algorithm compared with the traditional contrastive learning models such as SimCLR. The complexity of SimCLR is $O(N^{2}K^{2})$, while the complexity of SelfTime is $O(NK^{2})+O(NK)$, where $O(NK^{2})$ is the complexity of inter-sample relation reasoning module, and $O(NK)$ is the complexity of intra-temporal relation reasoning module. It can be seen that SimCLR scales quadratically in both training size $N$ and augmentation number $K$. However, in SelfTime, inter-sample relation reasoning module scales quadratically with the number of augmentations $K$, and linearly with the training size $N$, and intra-temporal relation reasoning module scales linearly with both augmentations and training size.
>
>
> 2. We would like to reclaim the contributions of our work here:
>
>     Our main contributions are three-fold: (1) we present a general self-supervised time series representation learning framework by investigating different levels of relations of time series data including inter-sample relation and intra-temporal relation. (2) We design a simple and effective intra-temporal relation sampling strategy to capture the underlying temporal patterns of time series. (3) We conduct extensive experiments on different categories of real-world time series data, and systematically study the impact of different data augmentation strategies and temporal relation sampling strategies on self-supervised learning of time series. By comparing with multiple state-of-the-art baselines, experimental results show that SelfTime builds new state-of-the-art on self-supervised time series representation learning.

---

> > ### Author Response · Authors · 2020-11-24
> > **Response #2**
> >
> > $\textbf{On video recognition experiments:}$
> >     Video data is a 3D visual data where the first two dimension includes spacial image information and the third dimension includes temporal information. However, there are little works that use video data for general time series representation learning [1-5], because video data contains more visual feature in each frame (time point), and requires a specific design of feature extraction network, while time series is far different structural data with less raw features at each time point. As same with those previous studies, we select classification task on general time series data as our basic task, and use linear evaluation [6, 7], domain transform [7], and embedding visualization [6, 7] to evaluate the performance of the proposed method.
> >
> >     [1] Ye, Lexiang, and Eamonn Keogh. "Time series shapelets: a new primitive for data mining." Proceedings of the 15th ACM SIGKDD international conference on Knowledge discovery and data mining. 2009.
> >
> >     [2] Franceschi, Jean-Yves, Aymeric Dieuleveut, and Martin Jaggi. "Unsupervised scalable representation learning for multivariate time series." Advances in Neural Information Processing Systems. 2019.
> >
> >     [3] Ma, Qianli, et al. "Learning representations for time series clustering." Advances in neural information processing systems. 2019.
> >
> >     [4] Cheng, Ziqiang, et al. "Time2Graph: Revisiting Time Series Modeling with Dynamic Shapelets." AAAI. 2020.
> >
> >     [5] Jawed, Shayan, Josif Grabocka, and Lars Schmidt-Thieme. "Self-supervised Learning for Semi-supervised Time Series Classification." Pacific-Asia Conference on Knowledge Discovery and Data Mining. Springer, Cham, 2020.
> >
> >     [6] Chen, Ting, et al. "A simple framework for contrastive learning of visual representations." ICML'20 (2020).
> >
> >     [7] Self-supervised relational reasoning for representation learning. Patacchiola and Storkey. NeurIPS'20

---

### Public Comment · ~Aaqib_Saeed1 · 2020-11-14
**Missing references**

As also pointed out by Reviewer 4, this paper entirely ignores following prior work as well:

[1] Saeed, Aaqib, Victor Ungureanu, and Beat Gfeller. "Sense and Learn: Self-Supervision for Omnipresent Sensors." arXiv preprint arXiv:2009.13233 (2020).

[2] Saeed, Aaqib, et al. "Multi-sensor data augmentation for robust sensing." 2020 International Conference on Omni-layer Intelligent Systems (COINS). IEEE.

[3] Saeed, Aaqib, et al. "Federated Self-Supervised Learning of Multi-Sensor Representations for Embedded Intelligence." IEEE Internet of Things Journal (2020).

[4] Sarkar, Pritam, and Ali Etemad. "Self-supervised ecg representation learning for emotion recognition." arXiv preprint arXiv:2002.03898 (2020).

---

> ### Author Response · Authors · 2020-11-24
> **Response**
>
> Thanks for the suggestions. We have added some missing related works from the recommendations.

---

### Author Response · Authors · 2020-11-24
**Overall response to the chairs and reviewers**


Thanks for the chairs and all reviewers! Overall, we have made the following improvements in the updated manuscript.

1. We have added more ablation study experiments on all datasets to illustrate the properties of the proposed method, and experimental results demonstrate the effectiveness of the proposed method.

2. We have added some missing related works and analyzed the difference between our work and those related works.

3. We have added new time-series representation learning baselines, which achieved state-of-the-art performance on the commonly used datasets, and experimental results demonstrate that our proposed method builds new state-of-the-art by outperforming those baselines with a large margin.

4. We rewrote some parts to improve the presentation and correct typo errors according to the reviewers' suggestions.


Please refer to the individual response for more details.

---

### Decision · Program_Chairs · 2021-01-07
**Final Decision**

**Decision:**

Reject

**Comment:**

This paper presents a general self-supervised time series representation learning framework. The organization is good, and the architecture is well motivated. However, the paper has limited novelty, and is a straightforward application of ideas in self-supervised learning literature.

Experimental results are not entirely convincing. The used dataset is small-scale that makes the task simple. A more thorough comparison with recent related work is needed. The presentation is also sometimes hard to follow.